# Genetic Variation in the *MBL2* Gene Is Associated with *Chlamydia trachomatis* Infection and Host Humoral Response to *Chlamydia trachomatis* Infection

**DOI:** 10.3390/ijms23169292

**Published:** 2022-08-18

**Authors:** Stephan P. Verweij, Remco P. H. Peters, Arnold Catsburg, Henry J. C. de Vries, Sander Ouburg, Servaas A. Morré

**Affiliations:** 1Department of Respiratory Medicine, University Medical Centre Utrecht, 3584 CX Utrecht, The Netherlands; 2Research Unit, Foundation for Professional Development, East London 5200, South Africa; 3Department of Medical Microbiology, University of Pretoria, Pretoria P.O. Box 14679, South Africa; 4MolGen B.V., 3905 NL Veenendaal, The Netherlands; 5Cluster of Infectious Diseases, Public Health Service Amsterdam, 1018 WT Amsterdam, The Netherlands; 6Center for Infection and Immunity Amsterdam (CINIMA), Department of Internal Medicine, Division of Infectious Diseases, Amsterdam University Medical Centers, 1105 AZ Amsterdam, The Netherlands; 7Department of Dermatology, Amsterdam University Medical Centers, 1105 AZ Amsterdam, The Netherlands; 8Centre for Infectious Diseases Control, National Institute for Public Health and the Environment (Rijksinstituut voor Volksgezondheid en Milieu, RIVM), 3720 BA Bilthoven, The Netherlands; 9Microbe & Lab B.V., 1105 AG Amsterdam, The Netherlands; 10Department of Genetics and Cell Biology, Faculty of Health, Medicine & Life Sciences, Institute of Public Health Genomics, Research Institute GROW, University of Maastricht, 6216 LK Maastricht, The Netherlands

**Keywords:** *Chlamydia trachomatis*, *MBL2* polymorphisms, IgG

## Abstract

This study aims to assess the potential association of *MBL2* gene single nucleotide polymorphisms (SNPs) to *Chlamydia trachomatis* infection. We analysed a selected sample of 492 DNA and serum specimens from Dutch Caucasian women. Women were categorized into four groups of infection status based on the results of DNA and antibody tests for *C. trachomatis*: Ct-DNA+/IgG+, Ct-DNA+/IgG−, Ct-DNA−/IgG+, and Ct-DNA−/IgG−. We compared six *MBL2* SNPs (−619G > C (*H*/*L*), −290G > C (*Y*/*X*), −66C > T (*P*/*Q*), +154C > T (*A*/*D*), +161A > G (*A*/*B*), and +170A > G (*A*/*C*)) and their respective haplotypes in relation to these different subgroups. The −619C (*L*) allele was less present within the Ct-DNA−/IgG+ group compared with the Ct-DNA−/IgG− group (OR = 0.49; 95% CI: 0.28–0.83), while the +170G (*C*) allele was observed more in the Ct-DNA+/IgG+ group as compared with the Ct-DNA−/IgG− group (OR = 2.4; 95% CI: 1.1–5.4). The *HYA*/*HYA* haplotype was more often present in the Ct-DNA−/IgG− group compared with the Ct-DNA+/IgG+ group (OR = 0.37; 95% CI: 0.16–0.87). The +170G (*C*) allele was associated with increased IgG production (*p* = 0.048) in *C. trachomatis* PCR-positive women. This study shows associations for MBL in immune reactions to *C. trachomatis*. We showed clear associations between *MBL2* genotypes, haplotypes, and individuals’ stages of *C. trachomatis* DNA and IgG positivity.

## 1. Introduction

*Chlamydia trachomatis* is the most prevalent bacterial sexually transmitted infection (STI) worldwide. The course of infection is variable; a recently acquired infection can have an active course, may be asymptomatic, or can be self-limiting. Untreated infections have a severe impact on the health of a patient owing to the possible development of late complications such as pelvic inflammatory disease, tubal pathology, and ectopic pregnancy [1].

The innate immune response is, similar to any other infectious disease, the first line of defence against infection by *C. trachomatis* [2]. Mannose-binding lectin (MBL) is an acute phase protein produced by the liver and has a central role in the innate immune response; MBL recognises and binds to patterns of glycoproteins present on microorganisms enabling opsonisation [3]. This C-type serum lectin binds to the 40 kDa major outer membrane protein of *C. trachomatis*, hampering invasion and infection of the host cell [4]. After binding of MBL to the outer member protein, a conformational change in mannose associated serine proteases (MASP-1 and MASP-2) occurs, which activates the lectin pathway of the complement system [5]. The MASP proteases cleave C4 and C2, generating C4b2a complexes that have C3 convertase activity [6,7]. MBL functions in close proximity to immunoglobulins (Ig) and facilitates opsonisation by macrophages [8].

The gene encoding the human MBL protein, *MBL2*, is located on chromosome 10 (10q11.2-q21). This *MBL2* gene incorporates four exons encoding a cysteine-rich region as well as a glycine-rich, collagen-like region (exon 1 and 2); a “neck” region (exon 3); and a carbohydrate-binding domain (exon 4) [9,10]. Six single-nucleotide polymorphisms (SNPs) of *MBL2* have been described [11]; that is, two SNPs in the promoter region: −619 G > C (*H/L*), −290 G > C (*Y*/*X*); one SNP at the 5’ untranslated region within the leader sequence: −66 C > T (*P*/*Q*); and three SNPs in exon 1: +154 C > T (*A*/*D*), +161 A > G (*A*/*B*), and +170 A > G (*A*/*C*). Figure 1 shows the rs-numbers and the relative positions of the SNPs on the gene. The promoter alleles are in strong linkage disequilibrium with the SNPs in exon 1, which generally results in seven haplotypes: *HYPA*, *LYPA*, *LYQA*, *LXPA*, *HYPD*, *LYPB,* and *LYQC* [12]. The fourth position in the haplotype, A, B, C, or D, is a combination of the three polymorphisms in exon 1, with A being the wild-type allele in all three positions and B, C, or D representing an SNP in the corresponding locus.

There is a strong association between the genotype of the *MBL2* gene and the level of MBL protein production. The haplotypes *HYPA* and *LYQA* are associated with high concentrations of MBL; *LYPA* and *LXPA* with intermediate/low concentrations; and *HYPD*, *LYQC*, and *LYPB* with MBL deficiency [11,13,14]. It has been shown that SNPs in exon 1 reduce the functionality of the protein and decrease MBL concentrations, thereby greatly reducing its complement-activating ability [15,16]. Serum MBL concentration is a determinant of susceptibility to infectious diseases and of disease outcome, and shows a strong correlation with allelic variants of the gene [11,13,17,18]. For example, it has been shown that children with exon 1 variants of the *MBL2* gene were more susceptible to meningococcal disease than children with wildtype alleles [19].

Owing to the effect of *MBL2* polymorphisms on susceptibility to infectious diseases, we aim to evaluate the role of these six known SNPs in the *MBL2* gene to a *C. trachomatis* infection. We will divide our study population into four biological subgroups and assess the role of these six SNPs, and we will determine the role of the SNPs in the production of IgG.

## 2. Results

### 2.1. Study Sample Characteristics

The median age of women in this analysis was 23 years (15–32 years). A total number of 65 (13%) samples were PCR positive, 73 (15%) samples were IgG positive, 139 (28%) samples were positive for both parameters, and 215 (44%) samples were negative for both *C. trachomatis* DNA and IgG. The median IgG response was 100 (50–1600). In total, 216 (44%) samples were obtained from women who reported STI-related symptoms and 159 (32%) had co-infection with other microorganisms upon inclusion; in particular, *Candida albicans* was prevalent (*n* = 144).

### 2.2. SNP Distribution

All genotypes analysed in this study were in Hardy–Weinberg equilibrium. Table 1 shows the overall SNP distribution and distribution of SNPs in relation to *C. trachomatis* infection. 

### 2.3. Association of SNPs and Stadium of Infection

We observed a significant difference in the carriage of the C (*L*) allele of the −619 SNP between the Ct-DNA−/IgG+ and Ct-DNA−/IgG− groups (*p* = 0.027). This observation was also shown in multivariate analysis for the Ct-DNA−/IgG+ group (OR: 1.5; 95% CI: 1.0–2.3; *p* = 0.036).

Distribution analysis between the Ct-DNA+/IgG− group and Ct-DNA−/IgG− group shows reduced carriage of the −290 C (*X*) allele in the Ct-DNA−/IgG− group (*p*_trend_ = 0.067).

Logistic regression analysis shows increased carriage of the +170 G (*C*) allele in the Ct-DNA+/IgG+ group compared with the Ct-DNA−/IgG− group, although this did not reach statistical significance (*p* = 0.06).

The SNP distribution did not differ significantly between the Ct-DNA−/IgG+ group and the Ct-DNA+/IgG− group. No significant differences in SNP distribution were observed when comparing the Ct-DNA+/IgG− group and the Ct-DNA+/IgG+ group.

No differences were observed when the Ct-DNA+/IgG− and Ct-DNA+/IgG+ groups were combined and compared to the Ct-DNA−/IgG+ group or Ct-DNA−/IgG− group, nor when these latter groups were combined in both univariate and multivariate analyses.

### 2.4. Effect of Homozygous and Heterozygous Carriage of Alleles and Susceptibility to Infection

Comparisons were made between homozygous and heterozygous carriage of alleles in order to assess susceptibility to infection. The following observations were made: the −619 C (*L*) allele was significantly more often present in the Ct-DNA−/IgG+ group compared with the Ct-DNA−/IgG− group (OR: 0.49, 95% CI: 0.28–0.83; *p* = 0.009). Another difference was observed comparing carriage of the +170 G (*C*) allele in the Ct-DNA+/IgG+ group compared with the Ct-DNA−/IgG− group (OR: 2.4, 95% CI: 1.1–5.4; *p* = 0.027).

In the multivariate analysis, carriage of the −619 C (*L*) allele was statistically associated with the Ct-DNA−/IgG− group compared with the Ct-DNA−/IgG+ group (aOR: 0.49; 95% CI: 0.28–0.83; *p* = 0.009). Moreover, for the Ct-DNA+/IgG+ group compared with the Ct-DNA−/IgG− group, carriage of the +170 G (*C*) allele was significantly associated with the Ct-DNA+/IgG+ infection group (OR: 2.4; 95% CI: 1.1–5.4; *p* = 0.031).

When comparing Ct-DNA+/IgG− to Ct-DNA+/IgG+, we see a statistical association for carriage of the +170 G (*C*) allele for the Ct-DNA+/IgG+ group (aOR: 4.1; 95% CI: 0.9–18.4; *p* = 0.048).

### 2.5. Haplotype Frequencies and Associations with Susceptibility

The haplotype distribution among all subgroups is summarized in Table 2. The haplotype *HYA/HYA* was more often present in the Ct-DNA−/IgG− group compared with the Ct-DNA+/IgG+ infection group (OR: 0.37, 95% CI: 0.16–0.87; *p* = 0.019). No other associations were observed for individual haplotype versus other haplotypes.

### 2.6. Association of SNPs with Immune Response

All PCR-positive women (*n* = 204) were included in this sub-analysis; 65 were IgG antibody negative and 139 were IgG positive. When comparing the distribution of SNPs with presence of IgG titre in *C. trachomatis*-positive women, carriage of the +170 G (*C*) allele was associated with presence of IgG in those *C. trachomatis*-positive women (*p* = 0.048, 3% vs. 12%), but median titre did not differ significantly between +170 AA (*AA*) and *G (**C*) alleles (median IgG titre +170 AA (*AA*): 100 (range 0–1600), median IgG titre +170 *G (**C*): 100 (range 0–400); Mann–Whitney U *p* = 0.32). No other associations for SNPs and IgG response were observed. Coinfection with other diagnosed microorganisms proved not to be a confounding factor.

## 3. Discussion

This study confirms an effect of polymorphisms in the *MBL2* gene in susceptibility to infection and the humoral IgG response to *C. trachomatis* infection. We divided our study population into four biological subgroups (Ct-DNA+/IgG+, Ct-DNA+/IgG−, Ct-DNA−/IgG+, and Ct-DNA−/IgG−) and have observed different associations between these subgroups. By introducing these subgroups, we were able to investigate the role of the *MBL2* SNPs in different possible stages of infection. It is known that genetic variants of the gene and associated variation in MBL concentration influence the susceptibility to and outcome of a wide variety of infectious diseases [17,19,20]. However, the associations described in this report are, especially with regard to the humoral response, the first for *C. trachomatis* infection.

The observed SNP frequency distribution was similar to that observed in other studies performed in Europe [12,14]. Our first observation was an association with carriage of the −619 C (*L*) allele and the Ct-DNA−/IgG+ group. This allele was significantly less common in the Ct-DNA−/IgG+ group compared with the Ct-DNA−/IgG− group. If we consider the different subgroups as stages of infection, this finding may indicate that carriage of the wildtype −619 GG (*HH*) genotype is involved in clearance of the infection, even though we do not find an association when comparing the SNP between the Ct-DNA−/IgG+ group and Ct-DNA+/IgG+ group. One can hypothesize that, within the Ct-DNA−/IgG+ group, there is a subgroup that had a reactive infection (Ct-DNA+/IgG+) and a subgroup that has cleared the infection spontaneously. Within the Ct-DNA−/IgG+ group, 34 patients (47%) out of 73 are carriers of the mutant −619 C (*L*) allele, and 39 patients (53%) out of 73 carry the −619 G (*H*) allele. We observed that 58% of the patients in the Ct-DNA+/IgG+ group were carriers of the mutant −619 C (*L*) allele. Hypothetically, 20 patients out of the 34 patients (59%) in the Ct-DNA+/IgG− group have had a reactive infection, whereas 14 (41%) cleared the infection without treatment. As such, we hypothesize that 36% of the total group of C (*L*) allele carriers in the Ct-DNA−/IgG+ group cleared the infection, which is plotted in Figure 2. By means of this method, we observe a theoretically larger association between patients who clear infection (within the Ct-DNA−/IgG+ group) and the Ct-DNA−/IgG− group (OR: 0.20, 95% CI: 0.10–0.39; *p* < 0.0001) in comparison with the association between the Ct-DNA−/IgG+ and Ct-DNA−/IgG− groups, which may explain why we did not find any association between the Ct-DNA−/IgG− group and the Ct-DNA+/IgG+ group.

We observed a difference in the distribution of the +170 G (*C*) allele; that is, the G (*C*) allele was more often present in women included in the Ct-DNA+/IgG+ group than those in the Ct-DNA−/IgG− group. The G (*C*) allele corresponds to an inadequate MBL and low MBL-producing haplotype. When combining the haplotypes corresponding to high-producing, low-producing, and deficient haplotypes, we did not observe any difference in distribution comparing the subgroups to the Ct-DNA−/IgG− group. This may be because of the division of ten haplotype combinations over the four subgroups, resulting in low haplotype frequencies per subgroup. Our third observation was the relation between the G (*C*) allele at position +170 and host IgG antibody production against *C. trachomatis*. This is relevant to know because increased susceptibility to *C. trachomatis* and prolonged infection may increase the chances of developing late complications [21]. Finally, we observed statistical trends in the Ct-DNA+/IgG− group and the Ct-DNA+/IgG+ group.

In vitro models support the role of *MBL2* in susceptibility to *C. trachomatis* infection. A previous in vitro study has shown the inhibitive role of MBL to *Chlamydia spp.* infections, including to *C. trachomatis* and *Chlamydia pneumoniae* infections, suggesting MBL has an influence on immunity to these infections [4]. Sziller et al. [22] observed, in a group of Hungarian women with proven tubal infertility, a significantly higher frequency of the *B* allele (+161 G) than in healthy controls. They hypothesize that a defect in first-line defence due to the polymorphism in *MBL2* contributes to persistence of the bacterium, which leads to damage of the Fallopian tubes. We do not find any association with this allele. This is not contradictory, as we did not assess the fertility of women in our analysis. It is possible that the role and/or mechanism of MBL is different for acquiring the infection than in complicating the course of disease. Two studies performed by Laisk and colleagues [23,24] investigated haplotypes in relation to *C. trachomatis*-induced tubal factor infertility (TFI) and observed that low-producing haplotypes (*LXA*/*LXA*, *HYA*/*O,* and *LYA*/*O*) were risk factors for developing TFI. They did not discover any association between MBL-deficient haplotypes or the very-low-producing haplotypes and TFI.

A relation between MBL deficiency and IgG has been observed previously by Roos et al. [25]. They pre-incubated mannose-coated plates with purified IgG or IgM antibodies and measured C4 deposition, a complement protein, upon addition of MBL-deficient serum. C4 concentrations were similar to that of MBL-sufficient serum, indicating a restorative role of antibodies for complement activation. This immunological redundancy may explain the variable inter-individual clinical outcome of disease in MBL-deficient persons, and has been proposed previously [26,27].

Carriage of an SNP in exon 1 reduces the functionality of MBL and additionally decreases MBL serum concentrations [15,16]. We have shown that patients carrying the G (*C*) allele at +170 were more likely to produce IgG antibodies than patients who had well-functioning MBL. Although this study has a relatively low number of patients with this allele, we believe that the observation is biologically plausible as the structural defect in MBL can be compensated for in vitro [28]. Laisk et al. [24] have also observed that the high-producing haplotype *HYA*/*HYA* was associated with TFI independent of *C. trachomatis* infection, but they could not confirm their previous results [23] when analysing *MBL2* genotypes and *C. trachomatis*-induced TFI [24]. We also find an association with this haplotype, albeit a protective one. Taking these results together, it seems that a high-producing haplotype is protective for *C. trachomatis* infection because of its complement activating ability, but also increases the risk of tubal pathology. This is an indication that MBL is important in immunity for *C. trachomatis*, but needs to be tightly regulated to prevent collateral damage.

The strengths of this study are that we used clearly defined subgroups on the basis of the presence of *C. trachomatis* DNA and/or specific IgG serum titres. We show associations of SNPs in different stadia of the infection, which indicates the significant immunological role of MBL to *C. trachomatis* infection, and may possibly have an effect on the clinical outcome.

This study has several limitations. Despite the sample size, the frequency of variants of exon 1 is relatively low, making it difficult to link susceptibility to *C. trachomatis* to one of these mutations. Additionally, their impact at the population level is expected to be limited because of the low frequency of these exon 1 SNPs in this population [29]. Furthermore, data on other confounding factors, such as *Mycoplasma genitalum* infection, bacterial vaginosis, birth control, and *C. trachomatis* virulence and load, are not available and may have influenced the results.

Creating biological subgroups was preferred over combining individuals based on parameters defining *C. trachomatis* infection, despite its potential limitations. For example, it is unknown when the individuals from the Ct-DNA−/IgG+ group were infected with *C. trachomatis*, or whether individuals from the Ct-DNA−/IgG− group actually had exposure. Moreover, not every infected individual will generate an antibody response, so this may have introduced some bias.

More research should be performed to assess the role of MBL to susceptibility to *C. trachomatis*. Larger, prospective studies should be conducted to gain further insight into the hypothesized immunological redundancy of antibodies and its effect on infection when an individual has an *MBL2* genotype coding for low or deficient MBL. Although an association between MBL and IgG production has not been observed, studies assessing this can be of interest because *C. trachomatis* serovars can induce different serological responses [30,31].

The results obtained from immunogenetic studies such as this one are of high relevance for public health and healthcare in general. Our results contribute to the understanding of disease pathogenesis of *C. trachomatis* infection, and our findings provide new insights into the immunological pathways that may contribute to the variable clinical course of the infection. Studies investigating SNPs in, for example, interleukin pathways are of similar importance and enhance the knowledge of chlamydial infection. Furthermore, our results may be integrated with existing immunogenetic knowledge, possibly aiding in the development of targeted and personalized approaches in the prevention, diagnosis, and treatment of the infection [28,32,33].

To conclude, this study suggests a role for MBL in immunological response to *C. trachomatis*. We observed associations of *MBL2* genotype and the stage of infection, and a clue for possible immunological redundancy was observed.

## 4. Materials and Methods

### 4.1. Sample Collection

A total of 492 samples were randomly selected from a previous case-control study [20]. Those samples were obtained from Dutch Caucasian women (age 15–35 years old) attending the STI outpatient clinic in Amsterdam, the Netherlands. Ethnicity (Dutch Caucasian) was self-reported and by means of questionnaires, which has been shown to be highly valid and representative in this context [34]. Upon inclusion, cervical swabs and serum samples were obtained from these women. Moreover, demographical, clinical, and laboratory data were available including age, symptoms, presence of coinfection systematically tested (including *Candida albicans*, *Neisseria gonorrhoeae*, and *Trichomonas vaginalis*), results of PCR test for *C. trachomatis* (COBAS Amplicor)*,* and the *C. trachomatis*-specific serum IgG titre (medac Diagnostika).

### 4.2. Ethical Approval

The Medical Research Involving Human Subjects Act (WMO, Dutch Law) stating official approval of the study by the Medical Ethical Committee does not apply to our collected anonymous human material (MEC Letter reference: #10.17.0046). The previous case-control study [35], from which the samples in this study were selected, was approved by the University’s Medical Ethical Committee. All participants of that study provided informed consent to use their samples anonymously for future research.

### 4.3. Laboratory Tests

The methods used for the detection and extraction of *C. trachomatis* DNA and determination of IgG serum titres have been described elsewhere [35]. In short, *C. trachomatis* detection was performed with COBAS Amplicor (Hoffman—La Roche, Basel, Switzerland) from DNA extracted from the cervical swab and was determined positive with when the Ct value was below 40. DNA for the MBL genotyping assay was extracted from peripheral blood mononuclear cells (PBMCs) by means of the isopropanol isolation method; a mixture of PBMC in PBS, nuclisense lysisbuffer, and glycogen was incubated for 30 min at 65 degrees Celsius and left to cool at room temperature. Isopropanol was added to the mixture and the samples were centrifuged. The supernatant was discarded and the remaining pellets were washed twice with 75% EtOH. The pellets were dissolved in T10 and stored for later analysis. The presence of IgG was determined with a *C. trachomatis*-specific ELISA (medac Diagnostika, Hamburg, Germany). Samples with a titre of ≥1:50 were considered positive.

### 4.4. Genotype Analysis

The following polymorphisms were analyzed in this study: −619 G > C (*L*/*H*), −290 G > C (*Y*/*X*), −66 C > T (*P*/*Q*), +154 C > T (*A*/*D*), +161 A > G (*A*/*B*), and +170 A > G (*A*/*C*). Genotyping of these six SNPs was performed as described elsewhere for a different context, i.e., pre-term children and risk for nocosomial infections [29,36]. Real-time PCR with four primer pairs and six probes was performed to determine the various alleles. The PCR conditions were 2 min at 50 °C, 10 min at 95 °C, and 40 cycles of 15 s at 95 °C and 1 min at 60 °C [22]. Haplotypes were inferred with PHASE v2.1.1 and SNPHAP [37,38,39].

### 4.5. Subgroups for Analyses

We classified the patients in this analysis into four subgroups to assess susceptibility for *C. trachomatis*. The first group includes women with both positive PCR and IgG determinations (Ct-DNA+/IgG+). The second subgroup includes women with a positive *C. trachomatis* PCR and negative IgG titre (Ct-DNA+/IgG−). The third subgroup includes women with a positive *C. trachomatis*-specific IgG titre and negative PCR result. The fourth subgroup represents women with both negative PCR and a negative *C. trachomatis* IgG titre (Ct-DNA−/IgG−). Figure 3 is a flowchart showing the inclusion criteria and subgroups.

To assess the role of MBL genotypes and haplotypes in initiating a humoral immune response, the following analysis was conducted: genotype and haplotype distributions were compared among women with a positive PCR result with and without IgG response to assess the potential associations of *MBL2* genotypes and haplotypes and serum IgG response to *C. trachomatis* infection. The classification by Steffensen et al. was used to correlate MBL haplotypes to MBL serum concentrations. In this classification, the haplotypes *HYA*/*HYA*, *HYA*/*LYA*, *LYA*/*LYA*, *HYA*/*LXA,* and *LYA*/*LXA* are considered to be associated with high MBL serum concentrations; the haplotypes *LXA*/*LXA*, *HYA*/*O,* and *LYA*/*O* correlate with low MBL concentrations; and *LXA*/*O* and *O*/*O* correlate with MBL deficiency. Allele *O* in this context is any promoter combination with any mutant allele of exon 1, whereas *A* in this context is any promoter combination with the wildtype alleles of the three exon 1 SNPs. The role of the *P*/*Q* allele is limited for MBL concentrations, so we did not include it in our haplotype [13].

### 4.6. Statistical Analyses

Descriptive statistics are provided and presented as the number (%) and median (range). All SNPs were assessed for Hardy–Weinberg equilibrium to test for deviation of Mendelian inheritance. Cross-tabulation of SNPs and haplotypes in women with and without *C. trachomatis* infection was performed, including χ^2^ statistics. Forward conditional multivariate regression analysis including all SNPs was used to observe associations found in univariate analysis. Finally, χ^2^ statistics and multivariate regression analyses were performed to assess the relation between the individual SNPs and dichotomised IgG production. Owing to the limited number of SNPs, correction for multiple testing was not performed to prevent underestimation of possible associations [40]. Analyses were performed using SPSS 13.0 (SPSS Inc., Chicago, IL, USA). A *p*-value of less than 0.05 was considered statistically significant, whereas 0.05 < *p* < 0.07 was considered a statistical trend.

## Figures and Tables

**Figure 1 ijms-23-09292-f001:**
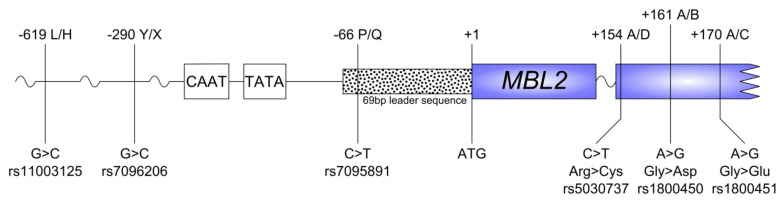
SNPs of the *MBL2* gene, relative positions of the SNPs from the translation site, and rs numbers.

**Figure 2 ijms-23-09292-f002:**
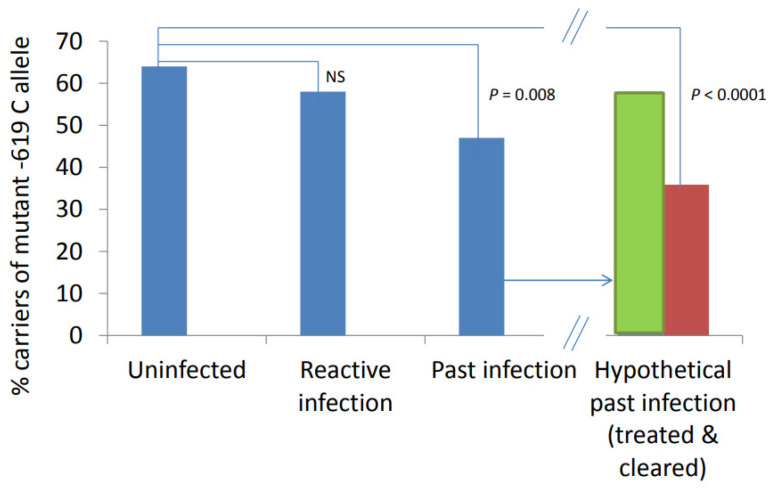
Percentage carriers of the mutant C (L) allele at position −619 of the Ct-DNA−/IgG, Ct-DNA+/IgG−, and Ct-DNA−/IgG+ groups. The Ct-DNA−/IgG+ group can be theoretically subdivided into patients having had a reactive infection in the past that has been treated (green bar), and patients having cleared the infection without treatment (red bar). This is represented in the fourth position of the figure. Patients clearing the infection (red bar) have a frequency of the C (L) allele significantly less often than the Ct-DNA−/IgG− individuals (OR: 0.20, 95% CI: 0.10-0.39; *p* < 0.0001).

**Figure 3 ijms-23-09292-f003:**
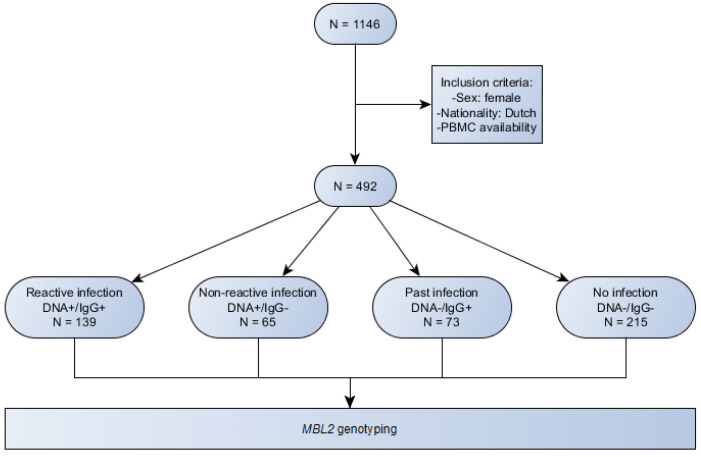
Flowchart showing the inclusion criteria and subgroups.

**Table 1 ijms-23-09292-t001:** SNP distributions at three stages of *Chlamydia* trachomatis infection.

			*Chlamydia trachomatis* Infection			
		Overall SNP Distribution	Ct-DNA+/IgG+	Ct-DNA+/IgG−	Ct-DNA−/IgG+	Ct-DNA−/IgG−
		*n* = 492 (%)	*n* = 139 (%)	*n* = 65 (%)	*n* = 73 (%)	*n* = 215 (%)
−619 G > C	GG	203 (41)	59 (42)	28 (43)	39 (53) *	77 (36)
	GC	224 (46)	67 (48)	30 (46)	24 (33)	103 (48)
	CC	65 (13)	13 (9.4)	7 (11)	10 (14)	35 (16)
−290 G > C	GG	301 (61)	88 (63)	35 (54)	41 (56)	137 (64)
	GC	167 (34)	45 (32)	29 (45)	27 (37)	66 (31)
	CC	24 (4.9)	6 (4.3)	1 (1.5)	5 (6.8)	12 (5.6)
−66 C > T	CC	280 (57)	76 (55)	36 (55)	41 (56)	127 (59)
	CT	187 (38)	51 (37)	27 (42)	28 (38)	81 (38)
	TT	25 (5.1)	12 (8.6)	2 (3.1)	4 (5.5)	7 (3.3)
+154 C > T	CC	430 (87)	122 (88)	58 (89)	63 (86)	187 (87)
	CT	60 (12)	15 (11)	7 (11)	10 (14)	28 (13)
	TT	2 (0.4)	2 (1.4)	0 (0.0)	0 (0.0)	0 (0.0)
+161 A > G	AA	386 (78)	108 (78)	49 (75)	57 (78)	172 (80)
	AG	99 (20)	28 (20)	15 (23)	15 (21)	41 (19)
	GG	7 (1.4)	3 (2.2)	1 (1.5)	1 (1.4)	2 (0.9)
+170 A > G	AA	459 (93)	123 (89)	63 (97)	69 (95)	204 (95)
	AG	32 (7)	15 (11)	2 (3.1)	4 (5.5)	11 (5.1)
	GG	1 (0.2)	1 (0.7)	0 (0.0)	0 (0.0)	0 (0.0)

* *p* < 0.05 (past infection vs. no infection).

**Table 2 ijms-23-09292-t002:** Haplotype frequencies for the three stages of *Chlamydia trachomatis* infection.

		Ct-DNA+/IgG+	Ct-DNA+/IgG−	Ct-DNA−/IgG+	Ct-DNA−/IgG−
	*MBL Production*	*n* = 139 (%)	*n* = 65 (%)	*n* = 73 (%)	*n* = 215 (%)
*HYA*/*HYA*	High	7 (5.0) *	4 (6.2)	6 (8.2)	27 (13)
*HYA*/*LYA*		22 (16)	12 (19)	8 (11)	37 (17)
*HYA*/*LXA*		19 (14)	8 (12)	5 (6.8)	26 (12)
*LYA*/*LYA*		10 (7.2)	2 (3.1)	8 (11)	13 (6.0)
*LYA*/*LXA*		14 (10)	12 (19)	13 (18)	23 (11)
*LXA*/*LXA*	Low	6 (4.3)	1 (1.5)	5 (6.8)	12 (5.6)
*HYA*/*O*		19 (14)	9 (14)	9 (12)	28 (13)
*LYA*/*O*		21 (15)	7 (11)	7 (9.6)	25 (12)
*LXA*/*O*	Deficient	12 (8.6)	9 (14)	9 (12)	17 (7.9)
*O*/*O*		9 (6.5)	1 (1.5)	3 (4.1)	7 (3.3)

* OR: 0.37, 95% CI: 0.16–0.87; *p* = 0.019 (Ct-DNA+/IgG+ vs. Ct-DNA−/IgG−).

## Data Availability

Data can be provided on reasonable request.

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
