# Peer review of "Genetic Variation in the MBL2 Gene Is Associated with Chlamydia trachomatis Infection and Host Humoral Response to Chlamydia trachomatis Infection"

_ijms, 2022, doi:10.3390/ijms23169292_

Round 1

Reviewer 1 Report

The manuscript Genetic variation in the MBL2 gene is associated with Chlamydia trachomatis infection and host humoral response to Chlamydia trachomatis infection has a sufficient number of samples included in the study. The methods are adequately described and used, results supported by the conclusions. It is stated by the authors that this is the only study of correlation between Chlamydia infection and MBL gene polymorphisms so I encourage the authors to include in the Introduction and Discussion some other variations in the immune system in general that can be correlated with Chlamydia infection. It would definitely improve the quality of the manuscript and help reduce the number of self-citations.

Author Response

The manuscript Genetic variation in the MBL2 gene is associated with Chlamydia trachomatis infection and host humoral response to Chlamydia trachomatis infection has a sufficient number of samples included in the study. The methods are adequately described and used, results supported by the conclusions. It is stated by the authors that this is the only study of correlation between Chlamydia infection and MBL gene polymorphisms so I encourage the authors to include in the Introduction and Discussion some other variations in the immune system in general that can be correlated with Chlamydia infection. It would definitely improve the quality of the manuscript and help reduce the number of self-citations.

We thank the reviewer for carefully reviewing our manuscript. Several different pathways have been described by us and other research groups concerning Chlamydia trachomatis infection. The MBL2 polymorphisms have, however, not been described yet in this context. We added additional information on associated SNPs within the discussion.

Reviewer 2 Report

The study conducted by the authors is interesting to the Chlamydia scientific community. This reviewer has few concerns:

1. What is the threshold Ct value for considering the samples Chlamydia positive?

2. Please elaborate section 2.3 of materials and method. 

Author Response

The study conducted by the authors is interesting to the Chlamydia scientific community. This reviewer has few concerns:

  1. What is the threshold Ct value for considering the samples Chlamydia positive?

  1. Please elaborate section 2.3 of materials and method.

We also thank this reviewer for carefully reviewing our manuscript.

  1. These clinical samples are deemed positive when Ct-values <40, this is the cut-off used in the Netherlands. We added the Ct cut-off within the materials and methods section.

  1. We added extra details within this section.
